# SOCIAL-MAMBA: EFFICIENT HUMAN TRAJECTORY FORECASTING WITH STATE-SPACE MODELS

## ABSTRACT

Human trajectory forecasting is crucial for safe navigation in crowded environments, requiring models that balance accuracy with computational efficiency. Efficiently modeling social interactions is key to performance in dense crowds. Yet, most recent methods rely on attention mechanisms, which are effective at capturing complex dependencies, but incur quadratic computational costs that scale poorly with the growing number of neighbors. Recently, Selective State-Space Models have provided a linear-time alternative; however, their inherently sequential design is misaligned with the unstructured and dynamic nature of social interactions. To address this challenge, we propose Social-Mamba, a forecasting architecture that reformulates social interactions as structured sequential processes. At its core is the Cycle Mamba block, a novel module that enables continuous bidirectional information flow. Social-Mamba organizes agents on a semantically ordered egocentric grid and introduces social triplet factorization, which decomposes interactions into temporal, egocentric, and goal-centric scans. These are dynamically integrated through a learnable social gate and global scan to generate accurate and efficient trajectory predictions. Extensive experiments on five trajectory forecasting benchmarks show that Social-Mamba achieves state-of-the-art accuracy while offering superior parameter efficiency and computational scalability. Furthermore, embedding Social-Mamba into a flow-matching framework further enhances both accuracy and efficiency, establishing it as a flexible and robust foundation for future trajectory forecasting research.

## 1 INTRODUCTION

Human trajectory forecasting is essential for safe, real-time decision-making in applications ranging from autonomous robotics to sports analytics (Gao et al., 2024; Li et al., 2025). The key challenge lies in balancing predictive accuracy and computational efficiency in dynamic, multi-agent environments, where inaccurate or slow predictions can lead to unsafe outcomes.

A central factor in accurate forecasting is the ability to model complex social interactions. Early deep learning methods relied on LSTMs to encode agent histories, combined with social pooling to aggregate neighbor influences (Alahi et al., 2016). Graph Neural Networks (GNNs) introduced more structure by representing agents as nodes and enabling explicit message passing (Salzmann et al., 2020). More recently, Transformers have become the dominant paradigm, using self-attention (Vaswani et al., 2017) to capture global all-to-all interactions and delivering significant improvements (Yuan et al., 2021; Girgis et al., 2021b; Saadatnejad et al., 2023). However, the **quadratic cost of attention** (Vaswani et al., 2017) with respect to the number of agents, along with more parameters, makes these models prohibitively expensive in crowded scenes.

Recently, the research community has turned to more efficient architectures, State Space Models (SSMs), particularly Mamba (Gu & Dao, 2023), as a highly promising solution to relieve this computational barrier. With their linear-time complexity and proven success in modeling long-range dependencies in other domains, SSMs are theoretically well-suited for processing long temporal sequences in trajectory data. Several pioneering works have attempted to leverage these advantages (Capellera et al. (2025); Xu & Fu (2024); Huang et al. (2025) Zhang et al. (2024a)). However, these attempts have revealed a fundamental conflict between the native design of SSMs and the nature of multi-agent dynamics.

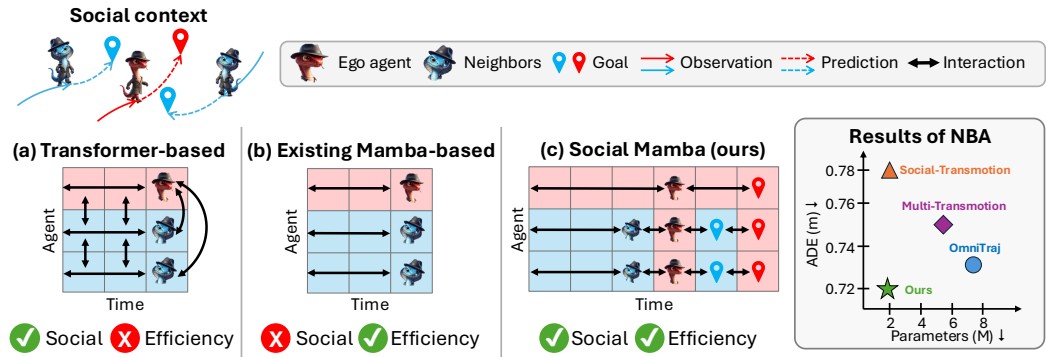

Figure 1: **Comparison of interaction modeling approaches.** (a) Transformers capture all-pair interactions but incur quadratic complexity. (b) Prior Mamba-based models are restricted to temporal-only modeling. (c) Our Social-Mamba integrates ego-agent context to enable efficient social interaction modeling, achieving state-of-the-art accuracy with up to 75% fewer parameters.

Directly applying SSMs to model social interactions exposes two critical and unresolved challenges. First, there is a **fundamental structural mismatch**: SSMs are designed for ordered, one-dimensional sequences, while social interactions are unstructured relationships in a 2D or 3D space. Forcing agents into an arbitrary sequence is not only unnatural but also risks destroying crucial spatial information and imposing an artificial hierarchy. Second, a significant **information flow mismatch exists**. Standard SSMs process information unidirectionally, yet social interactions are often holistic and bidirectional. The intent behind an agent's early action is often clarified only by what happens later, a retrospective context that a purely forward-pass model cannot capture.

In confronting these challenges, existing approaches have resorted to workarounds rather than fundamental solutions. Some models restrict Mamba to modeling only the temporal history of individual agents, sidestepping the social modeling problem entirely (Capellera et al., 2025; Xu & Fu, 2024). Others alter Mamba's core mechanism, reformulating it into an attention-like operator to handle the unstructured data (Huang et al., 2025), which sacrifices the unique properties of the original SSM architecture. Recently, MambaPTP (Zhang et al., 2025) pioneered the use of a pure Mamba architecture for trajectory prediction. However, it restricts social interaction modeling to the decoding stage and applies generic sequential scanning to neighbor embeddings, leaving the structural mismatch between 1D SSMs and unstructured social scenes unaddressed. Consequently, no prior work has successfully used an SSM in its native form for social interaction reasoning, leaving its full potential for trajectory forecasting unrealized.

To bridge this gap, we propose **Social-Mamba**, a novel architecture that adapts the sequential nature of SSMs to effectively model unstructured social dynamics. Our key idea is to integrate social interaction into conventional temporal scanning by imposing a meaningful structure on the scene and decomposing interactions into selective sequential scans. First, we introduce an **ego-centric social grid**, which organizes neighbors into a semantically meaningful sequence from the perspective of the ego agent, resolving the ordering problem. Next, we design a **social triplet factorization**, which replaces one complex scan with three distinct sequential scans: a temporal scan for individual agent histories, an ego-centric scan for neighbor influence on the ego's current state, and a goal-centric scan for how neighbors affect the ego's path toward its goal. Interaction is enabled by inserting the ego's state and goal tokens into neighbor sequences before scanning, allowing different interactions to be processed in a temporal manner. We show the main comparison in fig. 1. Finally, we perform **social fusion**, where context-dependent weights are learned for each interaction in the triplet, followed by a global scan to aggregate the information.

At the heart of these modules is the **Cycle Mamba** (CM) block, a novel bidirectional SSM designed for richer contextual understanding. In social scenarios, early actions are often reinterpreted by later events, and neighbor influence on the ego agent is inherently bidirectional. Standard bidirectional models struggle with this (Zhang et al., 2024b; Capellera et al., 2025; Xu & Fu, 2024), as their forward and backward passes remain disconnected. CM resolves this by concatenating the forward sequence with its reverse, ensuring uninterrupted hidden state flow. This allows the backward pass

to be explicitly conditioned on the forward context, mirroring human social reasoning. Furthermore, the single-pass design can save half of the parameters to improve memory efficiency.

In summary, Social-Mamba is the first trajectory forecasting architecture built entirely on Mamba, bridging the gap between sequential SSMs and unstructured social interactions. By introducing the CM, we further strengthen bidirectional continuity and save parameters. Extensive experiments demonstrate that Social-Mamba achieves state-of-the-art accuracy on five benchmark datasets while offering superior parameter efficiency and computational scalability.

## 2 RELATED WORK

### 2.1 HUMAN TRAJECTORY FORECASTING

Research in human trajectory forecasting has advanced along two main axes: *modeling social interactions* and *capturing the uncertainty of future paths*. For interaction modeling, early work introduced *social pooling* mechanisms that aggregated neighboring context in an unstructured way (Alahi et al., 2016; Kothari et al., 2021). To impose more explicit structure, GNNs emerged as a dominant paradigm, representing agents as nodes and enabling message passing for structured reasoning (Huang et al., 2019; Li et al., 2020; Salzmann et al., 2020; Xu et al., 2022a; Mohamed et al., 2020). More recently, *attention-based methods* have provided even greater flexibility, dynamically weighting all agents to capture global, context-dependent influences (Messaoud et al., 2020; Giuliari et al., 2021; Yuan et al., 2021; Girgis et al., 2021a). Orthogonal to interaction modeling is the challenge of multimodality, as multiple futures are often plausible. Addressing this requires generative approaches, which have evolved from GANs (Gupta et al., 2018; Sadeghian et al., 2019; Kosaraju et al., 2019) and VAEs (Schmerling et al., 2018; Ivanovic et al., 2020; Xu et al., 2022c) to more powerful diffusion-based techniques (Gu et al., 2022b; Mao et al., 2023; Bae et al., 2024) and, most recently, flow matching models (Fu et al., 2025), which generate higher-quality and more consistent trajectory distributions.

### 2.2 MAMBA FOR TRAJECTORY FORECASTING

Recent works have begun to adopt the Mamba architecture to improve temporal modeling in trajectory prediction. For example, U2Diff (Capellera et al., 2025) replaces the Transformer encoder with a bidirectional Mamba, enabling richer temporal dynamics without the need for positional encodings. Similarly, Sports-Traj (Xu & Fu, 2024) incorporates a bidirectional temporal Mamba within its Transformer encoder to capture temporal dependencies in sports data. In the autonomous driving domain, Tamba (Huang et al., 2025) leverages Mamba to redesign the encoder–decoder architecture, replacing self-attention with a more efficient linear-time alternative. Despite these advances, a key limitation remains: existing approaches restrict Mamba to temporal modeling and do not extend it to explicitly capture social interactions between agents. This leaves the potential of SSMs for multi-agent reasoning largely unexplored.

## 3 PRELIMINARIES

**Selective State Space Models**. Classical SSMs (Kalman, 1960) describe sequential data using hidden dynamics governed by linear systems of differential equations. Recently, structured SSMs such as S4 (Gu et al., 2021) showed that these models can be adapted for deep learning: by parameterizing the system matrices and discretizing with efficient rules such as zero-order hold (ZOH), S4 achieves linear-time sequence modeling while retaining strong long-range memory. This makes S4 a competitive alternative to quadratic-cost Transformers (Vaswani et al., 2017). However, S4 learns a time-invariant system, where parameters remain fixed across the input. To increase flexibility, Mamba (Gu & Dao, 2023) introduces selectivity: system parameters are conditioned on the input, allowing the model to adapt its dynamics over time. Formally, given input $x_t$, hidden state $h_t$, and output $y_t$, the continuous dynamics are:

$$h'(t) = \mathbf{A}(t)h(t) + \mathbf{B}(t)x(t), \quad y(t) = \mathbf{C}(t)h(t), \tag{1}$$

Figure 2: **Comparison of Bidirectional Mamba architectures.** (a) A conventional bidirectional Mamba uses two isolated passes, suffering from a contextual disconnect as information is only fused at the output layer. (b) CM performs a single, continuous scan. This ensures state continuity by having the forward pass directly initialize the backward pass, enabling a more integrated fusion with significantly fewer parameters.

where $\mathbf{A}(t)$ is the state matrix, $\mathbf{B}(t)$ is the input matrix, and $\mathbf{C}(t)$ is the output matrix, which can be discretized as:

$$h_t = \bar{\mathbf{A}}(t)h_{t-1} + \bar{\mathbf{B}}(t)x_t, \quad y_t = \mathbf{C}(t)h_t, \tag{2}$$

with $\bar{\mathbf{A}}(t)$ and $\bar{\mathbf{B}}(t)$ computed via ZOH. Selective SSMs differ from S4 by replacing the fixed parameters $(\Delta, \mathbf{B}, \mathbf{C})$ with input-dependent mappings:

$$\Delta(t) = \tau_\Delta(\Delta + s_\Delta(x_t)), \quad \mathbf{B}(t) = s_B(x_t), \quad \mathbf{C}(t) = s_C(x_t), \tag{3}$$

where $\tau_\Delta$ is a softplus function and $s_\Delta, s_B, s_C$ are learned projections. This selective mechanism enables the model to modulate how information flows through time, effectively bridging the efficiency of SSMs with the adaptability of Transformers.

**Ego-Centric human trajectory prediction.** The goal of ego-centric trajectory forecasting is to predict the future path of a specific ego agent, denoted as $e$, by considering its own motion history as well as the social context provided by all neighboring agents. Formally, consider $N$ agents observed for $T_{\text{obs}}$ time steps, where the trajectory of agent $i$ is $X_i = (x_i^1, x_i^2, \ldots, x_i^{T_{\text{obs}}})$, $x_i^t \in \mathbb{R}^2$. The set of observed trajectories for all agents is $\mathbf{X} = \{X_e, X_1, X_2, \ldots, X_N\}$. The task is to learn a mapping $f$ that takes the histories of all agents and predicts the future motion of only the ego agent $e$ over the next $T_{\text{pred}}$ time steps: $f : \mathbf{X} \mapsto \hat{Y}_e$, where the predicted trajectory $\hat{Y}_e = (\hat{y}_e^{T_{\text{obs}}+1}, \ldots, \hat{y}_e^{T_{\text{obs}}+T_{\text{pred}}})$, $\hat{y}_e^t \in \mathbb{R}^2$.

## 4 METHODOLOGY

We first introduce the Cycle Mamba block, the core module that enables continuous bidirectional information flow within our framework. Building on this foundation, we then present the complete Social-Mamba architecture, which integrates structured representations of social interactions with novel scanning strategies for trajectory forecasting.

### 4.1 CYCLE MAMBA BLOCK

A fundamental challenge in modeling sequential data, such as trajectories, is capturing context from both past and future elements. Conventional recurrent architectures typically address this through contextual isolation: two independent models process the forward and backward sequences separately, and their outputs are combined only at a final late fusion stage. While functional, this separation prevents the model from learning the seamless, continuous dependencies that exist between the two directions.

To overcome this limitation, we propose the CM, an architecture that unifies bidirectional processing into a single, continuous scan, as shown in fig. 2. By creating a **cycle** sequence, the model's internal

state propagates naturally from the backward context directly into the forward pass, enabling a more deeply integrated state-level fusion. The mechanism is formalized as follows:

**Sequence construction**: Given an input sequence of $L$ vectors, $S_{\text{fwd}} = (s_1, s_2, \ldots, s_L)$, where $s_t \in \mathbb{R}^D$, we first construct its reverse, $S_{\text{bwd}} = (s_L, s_{L-1}, \ldots, s_1)$. The cycle sequence, $S_{\text{cycle}} \in \mathbb{R}^{2L \times D}$, is formed by their concatenation:

$$S_{\text{cycle}} = [S_{\text{bwd}}; S_{\text{fwd}}] = (s_L, \ldots, s_1, s_1, \ldots, s_L). \tag{4}$$

**Continuous state-space scan**: A single Mamba model, defined by its state-space parameters $(\bar{A}, \bar{B}, C, D)$, processes this unified sequence. The hidden state $h_k \in \mathbb{R}^N$ evolves over the $2L$ timesteps according to the recurrence:

$$h_k = \bar{A}h_{k-1} + \bar{B}u_k, \quad y_k = Ch_k + Du_k, \tag{5}$$

where $u_k$ is the $k$-th vector of $S_{\text{cycle}}$. The complete output is $O_{\text{cycle}} = (o_1, \ldots, o_{2L})$.

**Output reconstruction**: The output sequence $O_{\text{cycle}}$ is deconstructed back into its backward and forward components. Let $O_{\text{bwd}} = (o_1, \ldots, o_L)$ and $O_{\text{fwd}} = (o_{L+1}, \ldots, o_{2L})$. The final output $O \in \mathbb{R}^{L \times D'}$ is obtained by aligning and merging these components, for example through element-wise addition:

$$O = O_{\text{fwd}} + \text{flip}(O_{\text{bwd}}). \tag{6}$$

The novelty of CM lies in its information flow. In a standard bidirectional model, the forward pass begins with a zero-initialized state $h_0^f = 0$, making it ignorant of the future. In our approach, the forward pass (processing $x_1$ at timestep $k = L + 1$) is initialized with the state $h_L$. This state is the final hidden state of the backward pass over $(x_L, \ldots, x_1)$. By unrolling the recurrence, we see that $h_L$ is a comprehensive summary of the entire reversed sequence:

$$h_L = \sum_{i=1}^{L} \bar{A}^{L-i} \bar{B} x_{L-i+1}. \tag{7}$$

Therefore, the very first step of the forward pass, computing the state for $x_1$, is:

$$h_{L+1} = \bar{A}h_L + \bar{B}x_1. \tag{8}$$

Equation 8 demonstrates that the forward pass is **explicitly conditioned on a compressed representation of the entire future context**. Instead of relying on a late fusion of two independent analyses, CM performs an **integrated, state-level fusion** where the forward representation is a direct, causal function of the backward one. This continuous propagation of the hidden state allows for a more sophisticated and cohesive model of bidirectional dependencies, which is especially critical for complex sequential tasks like trajectory forecasting.

## 4.2 SOCIAL-MAMBA

The Social-Mamba model is designed to learn hierarchical social representations through a multi-stage process. It begins by structuring the scene from an ego-centric viewpoint and then applies a series of specialized interaction modules, as shown in fig. 3.

**Social grid preparation.** We let the set of observed trajectories be $\mathbf{X} = \{X_e, X_1, ..., X_N\}$, where $X_i = (x_i^1, ..., x_i^{T_{obs}})$ and $x_i^t \in \mathbb{R}^2$. For a selected ego agent $e$, we define a permutation $\pi$ that sorts all agents $j \in \{1, ..., N\}$ based on their distance to agent $e$ at time $T_{obs}$:

$$\pi = \text{argsort}_j(||x_j^{T_{obs}} - x_e^{T_{obs}}||_2). \tag{9}$$

This ordering is used to construct an input tensor $\mathbf{S} \in \mathbb{R}^{N \times (T_{obs} + T_{pred}) \times 2}$, where the first dimension is ordered according to $\pi$. The future coordinates are initialized with zero vectors. This tensor is then encoded into a higher-dimensional representation, our initial social grid $\mathbf{Z}^{(0)} \in \mathbb{R}^{N \times T \times D}$, where $T = T_{obs} + T_{pred}$ and $D$ is the model dimension:

$$\mathbf{Z}^{(0)} = \text{MLP}(\mathbf{S}). \tag{10}$$

**Social triplet interaction.** We factorize social interactions into three parallel streams, where $\mathbf{Z}_i^{(0)} \in \mathbb{R}^{T \times D}$ is the sequence for agent $i$. We use $\text{CM}(\cdot)$ to denote the Cycle Mamba block.

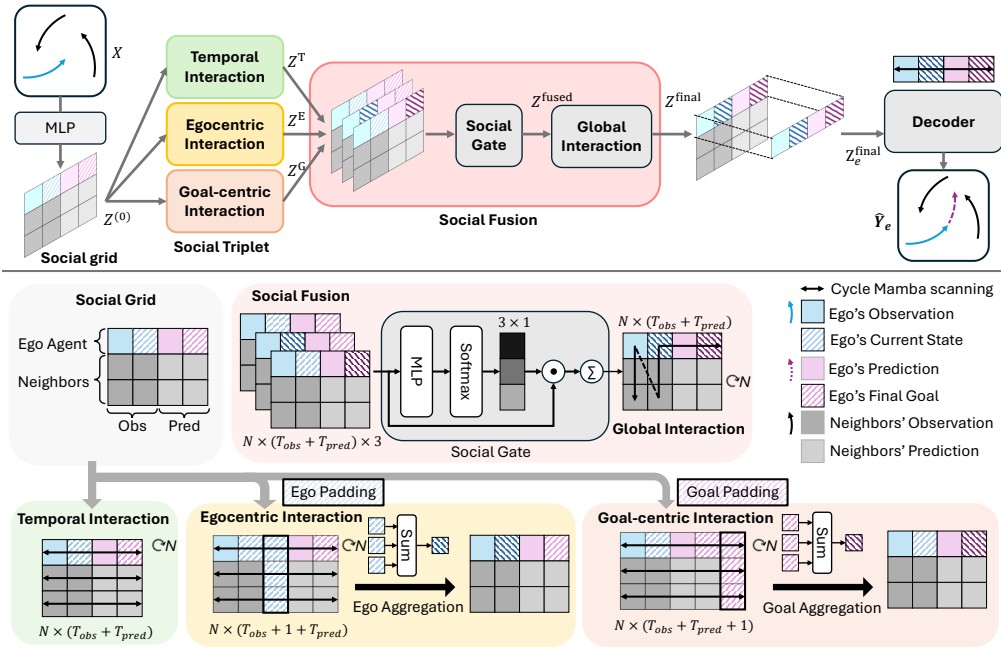

Figure 3: **Overview of the Social-Mamba framework**. The model first establishes an ego-centric view by creating a sorted social grid. This grid is then processed by three parallel interaction modules—temporal, egocentric, and goal-centric—which use our Cycle Mamba blocks to capture different facets of social influence. A dynamic gating network fuses these representations, and a final scan across agents captures global interactions to produce the ego-centric trajectory prediction.

- **Temporal interaction:** Each agent's sequence is processed independently to capture individual motion dynamics:

$$\mathbf{Z}_i^T = \mathrm{CM}(\mathbf{Z}_i^{(0)}). \tag{11}$$

- **Egocentric interaction:** To model the influence of the ego agent's current state, its token at $T_{obs}$, denoted $z_{e,T_{obs}}^{(0)}$, is inserted into each neighbor's sequence. Let $\mathbf{Z}'_j$ be the augmented sequence for neighbor $j$:

$$\mathbf{Z}'_j = [z_{j,1}^{(0)}, ..., z_{j,T_{obs}}^{(0)}, z_{e,T_{obs}}^{(0)}, z_{j,T_{obs}+1}^{(0)}, ..., z_{j,T}^{(0)}], \tag{12}$$

where $\mathbf{Z}'_j \in \mathbb{R}^{(T+1)\times D}$, and we use $\mathrm{CM}(\cdot)$ to scan along the time axis:

$$\mathbf{Z}'^E_j = \mathrm{CM}(\mathbf{Z}'_j), \quad z_{e,T_{obs}}^E = \sum_{t=T_{obs}}^{T_{obs}+1} w_t \cdot z'_{e,t} + \sum_{j=1}^{N} w_j \cdot z'_{j,T_{obs}+1}, \tag{13}$$

where $\mathbf{Z}_j'^E \in \mathbb{R}^{(T+1)\times D}$, and $\mathbf{Z}_j'^E = [z'_{j,1}, ..., z'_{j,T_{obs}}, z'_{e,T_{obs}}, z'_{j,T_{obs}+1}, ..., z'_{j,T}]$. The output is processed by learnable weighted sums and abandons the padding tokens to maintain the same shape that $\mathbf{Z}_j^E \in \mathbb{R}^{T\times D}$.

- **Goal-centric interaction:** Similarly, to model the influence of the ego's goal, its token at time $T$, $z_{e,T}^{(0)}$, is appended to each neighbor's sequence $\mathbf{Z}''_j$: $\mathbf{Z}''_j = [z_{j,1}^{(0)}, ..., z_{j,T}^{(0)}, z_{e,T}^{(0)}]$. We use $\mathrm{CM}(\cdot)$ to scan along the time axis:

$$\mathbf{Z}_j''^G = \mathrm{CM}(\mathbf{Z}''_j), \quad z_{e,T}^G = \sum_{t=T}^{T+1} w_t \cdot z''_{e,t} + \sum_{j=0}^{N} w_j \cdot z''_{j,T+1}, \tag{14}$$

where $\mathbf{Z}_j''^G \in \mathbb{R}^{(T+1)\times D}$, and $\mathbf{Z}_j'^E = [z''_{j,1}, ..., z''_{j,T_{obs}}, z''_{e,T_{obs}}, z''_{j,T_{obs}+1}, ..., z''_{j,T}]$. We use learnable weights to sum the goal tokens to merge information. To maintain the same shape, we remove the padding tokens that $\mathbf{Z}_j^G \in \mathbb{R}^{T\times D}$.

**Social fusion and decoding.** The three representations, $\mathbf{Z}^T, \mathbf{Z}^E, \mathbf{Z}^G$, are fused using a dynamic gating mechanism. First, they are concatenated, and a gating network computes weights for each interaction type:

$$\mathbf{W} = \text{Softmax}(\text{MLP}(\text{Concat}(\mathbf{Z}^T, \mathbf{Z}^E, \mathbf{Z}^G))), \tag{15}$$

where $\mathbf{W} = \{w_T, w_E, w_G\}$. The fused representation $\mathbf{Z}^{\text{fused}}$ is a weighted sum:

$$\mathbf{Z}^{\text{fused}} = w_T \odot \mathbf{Z}^T + w_E \odot \mathbf{Z}^E + w_G \odot \mathbf{Z}^G. \tag{16}$$

To model global interactions, we apply a Mamba scan along the agent axis (the first dimension of $\mathbf{Z}^{\text{fused}}$). Finally, the socially-aware representation for the ego agent, $\mathbf{Z}_e^{\text{final}}$, is isolated and passed to a final decoder network, a simple bidirectional Mamba and $K$ MLP projection heads, to predict the future trajectory:

$$\hat{Y}_e = \text{Decoder}(\mathbf{Z}_e^{\text{final}}), \tag{17}$$

where $\hat{Y}_e \in \mathbb{R}^{K \times T_{pred} \times 2}$ is the final predicted $K$ trajectories of the ego agent.

### 4.3 LOSS

To account for the inherent multimodality of human motion, our model predicts $K$ possible future trajectories. We train the model using a best-of-K loss strategy. Let $\hat{\mathbf{Y}}_e = \{\hat{Y}_{e,1}, ..., \hat{Y}_{e,K}\}$ be the set of $K$ predicted trajectories for the ego agent $e$, and let $Y_e$ be the corresponding ground truth trajectory. The loss for this sample is the Mean Squared Error (MSE) of the prediction that is closest to the ground truth:

$$\mathcal{L}_e = \min_{k \in \{1,...,K\}} \frac{1}{T_{pred}} \sum_{t=T_{obs}+1}^{T_{obs}+T_{pred}} ||\hat{y}_{e,k}^t - y_e^t||_2^2. \tag{18}$$

The final training objective is the average of this loss over all samples in a batch. This encourages the model to generate at least one plausible and accurate future path among its $K$ hypotheses.

## 5 EXPERIMENTS

### 5.1 DATASETS

**NBA.** The NBA dataset contains the trajectories of all 10 players and the ball during professional basketball games (Linou et al., 2016). The constant presence of a fixed number of agents makes it a unique testbed for analyzing complex group dynamics. We follow the standard setup of predicting 20 future frames (4.0s) from 10 past frames (2.0s). To ensure comprehensive evaluation, we adopt three widely used splits: (i) the full split (NBA-Full) introduced in (Gao et al., 2024), which leverages all data in NBA-LED (Mao et al., 2023), and (ii) scenario-specific splits focusing on rebounding and scoring plays (Xu et al., 2022c), (iii) the mini split (NBA-LED). Our main experiments are conducted on NBA-Full, with additional evaluations on the other splits

**Stanford Drone Dataset (SDD).** SDD captures real-world pedestrian dynamics from a bird's-eye view across a university campus (Robicquet et al., 2016). Its diverse trajectories and crowded interactions make it well-suited for evaluating social forecasting models. Following prior work (Xu et al., 2022c), we use 8 observed frames (3.2s) to predict the subsequent 12 frames (4.8s). All trajectories are processed in meters.

**JackRabbot Dataset and Benchmark (JRDB).** JRDB is a large-scale, egocentric dataset recorded from a mobile social robot navigating both indoor and outdoor environments (Martin-Martin et al., 2021). It features diverse social interactions between humans, robots, and static obstacles, making it particularly challenging. Following the standard protocol in (Fang et al., 2025), we predict 12 future frames (4.8s) based on 9 observed frames (3.6s), with trajectories sampled at 2.5 Hz.

### 5.2 METRICS

We evaluate our model using standard metrics for multimodal trajectory forecasting, where the model predicts $K$ possible future trajectories.

**Minimum Average Displacement Error (minADE$_K$):** This is the average L2 distance between the ground-truth trajectory and the closest of the $K$ predicted trajectories, calculated over all timesteps in the prediction horizon.

**Minimum Final Displacement Error (minFDE$_K$):** the L2 distance between the final ground-truth position and the endpoint of the closest predicted trajectory among the $K$ candidates.

For simplicity, we refer to these metrics as ADE and FDE throughout the paper. Unless otherwise noted, we set $K = 20$, and all reported errors are measured in meters.

## 5.3 Quantitative Results

Table 1: **Comparison with baseline models on NBA-Full.** Models marked with * are pretrained on large-scale trajectory datasets. Best results are highlighted in bold.

| Method | ADE ↓ | FDE ↓ | Parameters (M) ↓ | GFLOPs ↓ |
|---|---|---|---|---|
| Social-Transmotion (Saadatnejad et al., 2023) | 0.78 | 1.01 | 2.0 | 0.87 |
| Multi-Transmotion* (Gao et al., 2024) | 0.75 | 0.97 | 5.7 | 0.87 |
| OmniTraj* (Gao et al., 2025) | 0.73 | 0.94 | 7.5 | 1.45 |
| Social-Mamba (ours) | **0.72** | **0.92** | **1.9** | **0.66** |

**NBA-Full.** Table 1 reports results on the NBA dataset. Social-Mamba consistently outperforms all baselines in both ADE and FDE. Notably, several strong competitors such as Multi-Transmotion and OmniTraj rely on pretraining with large-scale external datasets before fine-tuning. In contrast, Social-Mamba achieves superior accuracy by training solely from scratch on the official dataset, highlighting both its data efficiency and the effectiveness of its architecture.

**NBA Scoring and Rebounding.** As shown in table 2, Social-Mamba also achieves state-of-the-art results on the specialized Scoring and Rebounding splits. The model shows a particularly strong advantage in the Scoring scenario, where coordinated, goal-directed behavior is critical. In this setting, Social-Mamba reduces ADE by 8.2% and FDE by 7.3% compared to prior work, highlighting its effectiveness at modeling structured group interactions.

**SDD.** We report all results in meters and compare against baselines that follow the same protocol. Reporting in meters provides a standardized and meaningful comparison, as pixel-based metrics are influenced by camera parameters and resolution, whereas metric distances are absolute. For completeness, we also provide pixel-based results in section A.2. As shown in table 3, Social-Mamba achieves performance comparable to the current state-of-the-art on this challenging benchmark.

**JRDB.** Results on the JRDB dataset are presented in table 4. Social-Mamba consistently outperforms prior methods across different prediction horizons, demonstrating both stability and robustness. In particular, our method achieves substantial gains, improving ADE by up to 13% and FDE by 8.7% over the previous state-of-the-art.

**Efficiency and flexibility.** A central motivation of our work is to improve the efficiency of trajectory forecasting models. As shown in table 1, Social-Mamba achieves state-of-the-art accuracy on the NBA dataset while requiring significantly fewer parameters and lower computational cost (GFLOPs) than competing methods. Compared to the prior state-of-the-art, our model reduces parameters by 75% and GFLOPs by 54%. This inherent efficiency also makes the architecture highly flexible. To demonstrate this, we integrated Social-Mamba as a drop-in replacement for the Transformer-based encoder in a state-of-the-art flow-matching framework (Fu et al., 2025). The implementation details are in section A.5. Results in table 5 show that this integration not only improves accuracy but does so with an encoder that is 2.3× smaller, highlighting Social-Mamba's practicality as a lightweight and general-purpose social interaction module.

## 5.4 Qualitative Results

To further illustrate performance, we visualize predictions on NBA-Full in comparison with Multi-Transmotion in fig. 4. Social-Mamba consistently produces more accurate trajectories across diverse scenarios. Additional qualitative results, including NBA and JRDB, are provided in section A.6.

Table 2: **Comparison with baseline models on NBA Rebounding and Scoring.**

| Method | Rebounding | Scoring |
|---|---|---|
| Trajectron++ (Salzmann et al., 2020) | 0.98/1.93 | 0.73/1.46 |
| BiTrap (Yao et al., 2021) | 0.83/1.72 | 0.74/1.49 |
| SGNet-ED (Wang et al., 2022) | 0.78/1.55 | 0.58/1.30 |
| Social-VAE (Xu et al., 2022c) | 0.72/1.37 | 0.64/1.17 |
| RNLS & CLLS (Qiu et al., 2025) | 0.65/1.20 | 0.61/1.09 |
| Social-Mamba (ours) | **0.63/1.18** | **0.56/1.01** |

Table 3: **Comparison with baseline models on SDD.**

| Method | ADE/FDE↓ |
|---|---|
| Trajectron++ (Salzmann et al., 2020) | 0.34/0.58 |
| BiTrap (Yao et al., 2021) | 0.32/0.57 |
| SGNet-ED (Wang et al., 2022) | 0.33/0.58 |
| Social-VAE (Xu et al., 2022c) | 0.27/0.39 |
| NMRF (Fang et al., 2025) | **0.25**/0.39 |
| Social-Mamba (ours) | **0.25/0.38** |

Table 4: **Comparison with baseline models on JRDB across different horizons.**

| Method | 1.2s | 2.4s | 3.6s | 4.8s (total) |
|---|---|---|---|---|
| LED (Mao et al., 2023) | 0.05/0.07 | 0.09/0.14 | 0.14/0.21 | 0.18/0.28 |
| NMRF (Fang et al., 2025) | **0.04/0.05** | 0.08/0.11 | 0.11/0.17 | 0.15/0.23 |
| Social-Mamba (ours) | **0.04/0.05** | **0.07/0.10** | **0.10/0.15** | **0.13/0.21** |

Table 5: **Quantitative results of MoFlow using different social encoders on NBA-LED.**

| MoFlow encoder | ADE/FDE ↓ | Parameters (M) ↓ |
|---|---|---|
| Transformer | 0.71/0.87 | 1.3 |
| Social-Mamba | **0.70/0.85** | **0.5** |

Table 6: **Ablation of the interaction modules**.

| Temporal | Ego | Goal | Global | ADE/FDE ↓ |
|---|---|---|---|---|
| ✓ | ✗ | ✗ | ✓ | 0.735/0.939 |
| ✓ | ✓ | ✗ | ✓ | 0.729/0.928 |
| ✓ | ✗ | ✓ | ✓ | 0.727/0.928 |
| ✓ | ✓ | ✓ | ✓ | **0.719/0.919** |

Table 7: **Comparison with different interaction blocks**.

| Interaction block | ADE/FDE ↓ | Parameters (M) ↓ | GFLOPs ↓ |
|---|---|---|---|
| MHSA | 0.741/0.947 | 2.3 | 0.80 |
| Bidirectional Mamba | 0.741/0.948 | 2.4 | **0.66** |
| Cycle Mamba (ours) | **0.719/0.919** | **1.9** | **0.66** |

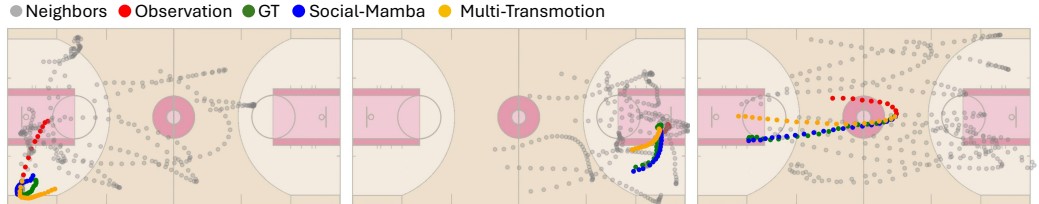

Figure 4: **Qualitative results of the NBA-Full**. Social-Mamba performs better in various scenarios, including abrupt turns, direction changes, and running to the other side of the field.

## 5.5 ABLATION STUDIES

We conduct ablations on NBA-Full to evaluate the design choices of Social-Mamba. Additional ablation study can be found in the section A.4.

**Impact of interaction modules**. We analyze the contribution of each interaction module. To preserve essential temporal and spatial reasoning, the temporal and global scans are kept as the baseline. As shown in table 6, removing either the goal-centric or ego-centric scan results in performance degradation, while combining both yields the best performance, highlighting their complementary roles.

**Interaction block.** We compare our CM block against other bidirectional or interaction-focused mechanisms, including a standard bidirectional Mamba (Zhang et al., 2024b; Xu & Fu, 2024; Capellera et al., 2025) and Multi-Head Self-Attention (MHSA) (Vaswani et al., 2017). As shown in table 7, CM achieves the best performance with the fewest parameters, demonstrating its effectiveness as an efficient building block.

## 6 CONCLUSION

We introduced Social-Mamba, the first forecasting architecture built entirely on the Mamba framework, to address the fundamental challenge of applying sequential SSMs to unstructured social dynamics. Our model bridges this gap by structuring the scene with an ego-centric social grid and

decomposing interactions via a social triplet factorization, powered by our novel bidirectional Cycle Mamba block. Across five benchmark datasets, Social-Mamba achieves new state-of-the-art performance with significantly greater computational efficiency and demonstrated flexibility, confirmed by its successful integration into a flow matching framework.

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

Table 8: **Comparison with baseline models on SDD.** The numbers are in pixels. Underlines denote the second best.

| Method | Venue | ADE/FDE $\downarrow$ |
|---|---|---|
| Trajectron++ (Salzmann et al., 2020) | ECCV'20 | 10.00/17.15 |
| BiTrap (Yao et al., 2021) | RAL'21 | 9.09/16.31 |
| MID (Gu et al., 2022a) | CVPR'22 | 9.08/17.04 |
| MemoNet (Xu et al., 2022b) | CVPR'22 | 8.56/12.66 |
| SGNet-ED (Wang et al., 2022) | RAL'22 | 9.69/17.01 |
| Social-VAE (Xu et al., 2022c) | ECCV'22 | 8.10/11.72 |
| LED (Mao et al., 2023) | CVPR'23 | 8.48/11.66 |
| TUTR (Shi et al., 2023) | ICCV'23 | 7.76/12.69 |
| ET-HighGraph (Kim et al., 2024) | CVPR'24 | 7.81/**11.09** |
| MoFlow-IMLE (Fu et al., 2025) | CVPR'25 | 7.85/12.86 |
| UniTraj (Xu & Fu, 2024) | ICLR'25 | 8.68/12.78 |
| NMRF (Fang et al., 2025) | ICLR'25 | **7.20**/ 11.29 |
| Social-Mamba (ours) | - | 7.54/11.74 |

## A    APPENDIX

The appendix provides additional details and analyses to complement the main paper. We begin with implementation details, including training configurations and architectural settings. Next, we present further experimental results, including pixel-based evaluation on SDD and performance on the deterministic TrajNet++ benchmark. We then conduct extended ablation studies, analyzing architectural design choices, fusion strategies, and decoder structures. To highlight the flexibility of our approach, we describe the integration of Social-Mamba into the MoFlow framework. Additional qualitative results are provided for NBA-Full and JRDB, including multimodal diversity visualizations and failure cases. Finally, we discuss limitations and future directions, followed by a statement on LLM usage in preparing this manuscript.

### A.1    IMPLEMENTATION DETAILS

Our model was trained on a single NVIDIA A100 GPU. The feature dimension of the social grid representation is set to 128. Within each Mamba block, the SSM state dimension is 16, and the convolution kernel size is 4. We trained the model for 100 epochs using the ADAM optimizer and a step-based learning rate schedule to improve convergence.

### A.2    ADDITIONAL EXPERIMENTAL RESULTS

For completeness, we report results on SDD in pixel-based coordinates in table 8. Social-Mamba achieves performance comparable to the state-of-the-art method NMRF. However, we argue that pixel-based evaluation is unreliable: the real-world distance represented by a pixel varies with camera perspective and calibration. This introduces inconsistencies that hinder fair comparison. We advocate reporting in meters as a standardized metric. As shown in table 3, Social-Mamba slightly outperforms NMRF under this evaluation. The discrepancy, the ranking of top methods changes between metrics, highlights the inconsistency of pixel-based evaluation and the reliability of real-world coordinates.

To further validate our model, we also evaluate Social-Mamba on a deterministic benchmark TrajNet++ (Kothari et al., 2021). As shown in table 9, Social-Mamba performs strongly under this deterministic setup, confirming its robustness across different evaluation protocols.

### A.3    EFFICIENCY ANALYSIS

To further validate the practicality of our approach, we provide additional evaluation of computational efficiency, including inference time, memory usage of models, and GPU cost during training. All experiments were conducted on a single NVIDIA A100 GPU, with a batch size of 1 during inference and 128 during training, following similar settings in (Zhang et al., 2025)

Table 9: **Comparison with baseline models on Trajnet++.** The ADE and FDE are deterministic.

| Method | ADE/FDE ↓ |
|---|---|
| DagNet (Monti et al., 2021) | 0.66/1.44 |
| Trajectron++ (Salzmann et al., 2020) | 0.55/1.16 |
| AutoBots (Girgis et al., 2021b) | 0.54/1.12 |
| Multi-Transmotion (Gao et al., 2024) | 0.54/1.13 |
| Social-Mamba (ours) | **0.53/1.10** |

Table 10: **Efficiency comparison of different interaction modules.**

| Model | Inference time (ms) | Model memory usage (MB) | Training memory usage (MB) |
|---|---|---|---|
| MHSA | 4.4 | 8.7 | 14356 |
| Bidirectional Mamba | 5.1 | 9.1 | 9422 |
| Cycle Mamba (ours) | **3.4** | **7.3** | **9375** |

Table 11: **Efficiency comparison of different forecasting models.**

| Model | Inference time (ms) | Model memory usage (MB) |
|---|---|---|
| Social-Transmotion | **1.8** | 7.6 |
| Multi-Transmotion | 7.3 | 21.83 |
| Social-Mamba (ours) | 3.4 | **7.3** |

**Interaction module analysis.** We first examine the efficiency of the proposed Cycle Mamba block compared to a standard Bidirectional Mamba and the MHSA. As shown in table 10, Cycle Mamba achieves the lowest memory footprint (7.3 MB) and the fastest inference time (3.4 ms). Notably, it outperforms the standard Bidirectional Mamba in speed, likely due to its continuous scan design, which avoids the overhead of managing two independent, disconnected passes. All modules maintain low resource consumption, confirming that the core building blocks of Social-Mamba are lightweight. In training, Cycle Mamba can be trained with the lowest GPU cost under the same batch size, demonstrating that our model has a lower hardware requirement during training.

**Model-level comparison.** We further extend this evaluation to the full model level, comparing Social-Mamba against recent Transformer-based baselines: Social-Transmotion (Saadatnejad et al., 2023) and Multi-Transmotion (Gao et al., 2024). The results in table 11 demonstrate Social-Mamba's superior scalability. While maintaining state-of-the-art accuracy, Social-Mamba operates with minimal resource consumption—requiring just 7.3 MB of memory and 3.4 ms inference time. This represents a significant efficiency gain over complex architectures like Multi-Transmotion, which requires over 7.3 ms per inference. It is worth noting that Social-Transmotion achieves a slightly faster inference time (1.8 ms); this is attributed to the Transformer's inherent advantage in parallel computation for short sequences, a behavior consistent with observations in (Zhang et al., 2025). Nevertheless, Social-Mamba provides the most favorable trade-off between high accuracy (as detailed in the main results) and low memory footprint, making it highly suitable for deployment in real-time systems.

### A.4 ADDITIONAL ABLATION STUDY

**Design choices of Social-Mamba.** We further analyze our architectural choices for the social triplet. Specifically, we compare our parallel design with a sequential variant where temporal, egocentric, and goal-centric interactions are applied in order. As shown in table 12, the parallel design achieves superior performance. We attribute this to our parallel fusion gate, which enables the model to dynamically adjust the contribution of each interaction type. In contrast, a sequential design risks information loss, as features from earlier interactions may be overwritten by subsequent ones.

We also investigate how to fuse the outputs of the social triplet (table 13). We compare our learnable softmax gating mechanism with a simple additive fusion that directly sums the three interaction terms without weights. The results demonstrate that incorporating learnable weights provides greater flexibility across scenarios. For instance, when the ego agent moves independently without

Table 12: **Comparison with sequential social triplet.**

| Social triplet | ADE/FDE $\downarrow$ |
|---|---|
| Sequential | 0.728/0.929 |
| Parallel | **0.719/0.919** |

Table 13: **Comparison of fusion strategies for the social triplet.**

| Social fusion | ADE/FDE $\downarrow$ |
|---|---|
| Addition | 0.744/0.954 |
| Learnable weights | **0.719/0.919** |

Table 14: **Comparison with MLP decoder.**

| Decoder | ADE/FDE $\downarrow$ |
|---|---|
| MLP | 0.730/0.931 |
| Mamba | **0.719/0.919** |

Table 15: **Comparison of different sorting strategie.**

| Strategy | ADE |
|---|---|
| Collision risk | 0.73 |
| Goal alignment | 0.73 |
| Proximity | 0.72 |
| Distance | 0.72 |

nearby interactions, the model can automatically downweight the influence of social context, while in crowded scenes it can emphasize neighbor interactions more strongly.

**Network structure.** We also analyze the core components of our network. First, we compare our Mamba-based decoder to a conventional MLP decoder (Saadatnejad et al., 2023; Gao et al., 2024) as shown in table 14. The results suggest that the Mamba decoder is more effective at generating high-quality trajectories.

**Social grid sorting strategies.** To examine the sensitivity of our model to the input order of neighbors, we evaluate performance under four different sorting heuristics:

1. **Collision risk:** sort neighbors by ascending Time-to-Collision (TTC). For approaching agents, TTC is approximated as $\tau = \frac{||p_{rel}||^2}{p_{rel} \cdot v_{rel}}$. Agents not on a collision course are assigned infinite TTC and sorted last.

2. **Goal alignment:** sort neighbors by descending cosine similarity between the neighbor's velocity $v_{neighbor}$ and the relative position vector pointing to the ego ($p_{ego} - p_{neighbor}$). This prioritizes neighbors explicitly moving toward the ego.

3. **Proximity:** apply a spatial filter that retains all agents within a relevance radius (e.g., $10m$) but imposes no specific order (random permutation). This isolates the impact of neighbor selection versus neighbor ordering.

4. **Distance (baseline):** sort neighbors by ascending Euclidean distance ($||p_{ego} - p_{neighbor}||$). This assumes immediate spatial proximity is the strongest proxy for interaction relevance.

Table 15 presents the ablation results on the NBA dataset, where all heuristics yield comparable ADE. This empirical evidence reveals that our model is largely **insensitive** to the specific order of neighbors. This robustness aligns with the design of our Social Triplet module, which aggregates interactions via summation, a process that is theoretically permutation-invariant. Consequently, the choice of sorting strategy is less about performance maximization and more about practical feasibility: the selected heuristic must rely on observable information in a real-world system, and the sorting logic must be consistently applied during both training and inference to maintain distribution alignment.

A.5 SOCIAL MAMBA WITH FLOW MATCHING

We integrated Social-Mamba into the MoFlow framework by replacing its Transformer-based context encoder with our model, and substituting the spatial-temporal Transformer with our global interaction scanning module. The motion decoder of MoFlow is left unchanged. The complete pipeline is illustrated in fig. 5.

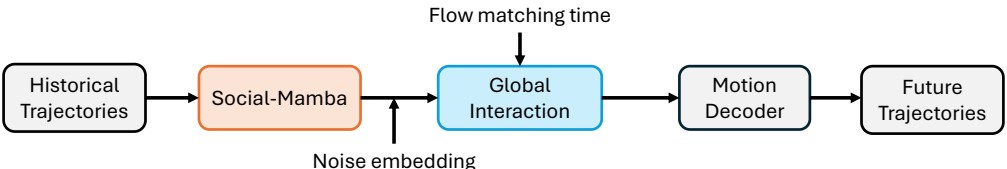

Figure 5: **Pipeline of MoFlow with Social Mamba**. We utilize Social-Mamba to model context and social-temporal interactions for MoFlow.

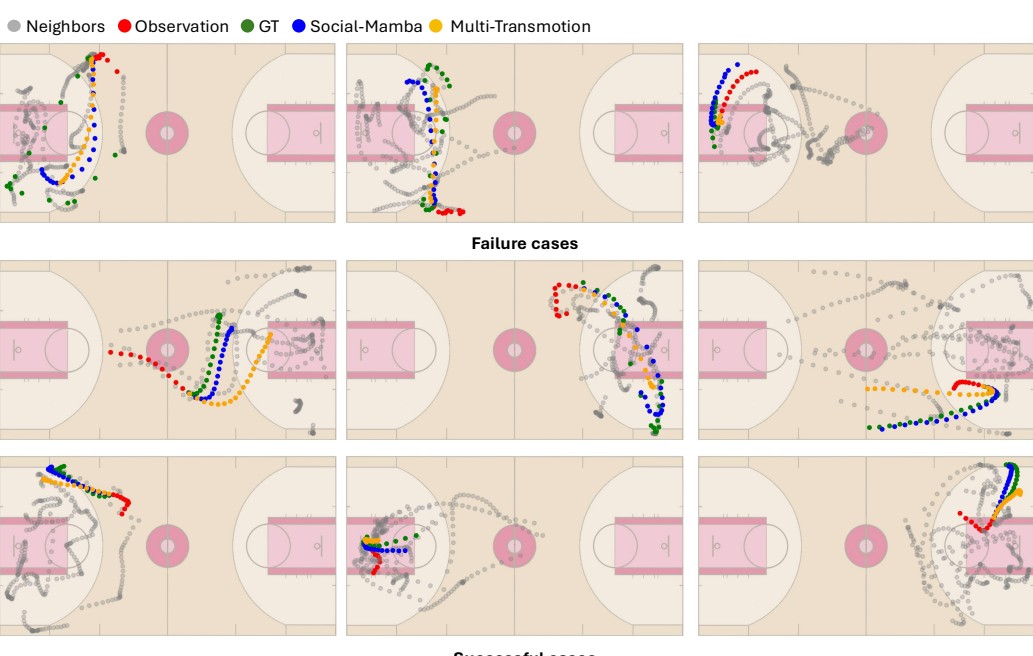

Figure 6: **Additional qualitative results of the NBA-Full**. Row 1 represents the failure cases. Rows 2 and 3 demonstrate successful cases.

### A.6 ADDITIONAL QUALITATIVE RESULTS

Figure 6 presents extra visualizations on NBA-Full. We include failure cases, where errors often occur during ball passes—events that induce abrupt velocity changes and irregular trajectories. Sudden turns also remain challenging. Nonetheless, Social-Mamba generally provides more accurate predictions than Multi-Transmotion.

Figure 7 visualizes the complete set of output modalities to assess prediction diversity. Social-Mamba generates trajectories that span plausible futures with realistic directions and velocities, showcasing its ability to capture multimodal behaviors. In (e), the play occurs in the paint during an attack, leading to more diverse predictions. We also note occasional boundary violations, as shown in (f).

In JRDB (Figure 8), we visualize crowded campus scenes. (a) shows Social-Mamba handling dense crowds, (b) demonstrates accurate predictions of highly non-linear behaviors, and (c) highlights a case with small deviations, where the prediction still avoids potential collisions.

### A.7 LIMITATIONS AND FUTURE WORK

While Social-Mamba demonstrates strong performance, several avenues remain open. First, our social ordering relies on ego-centric distance. Though effective, this does not capture social nuances

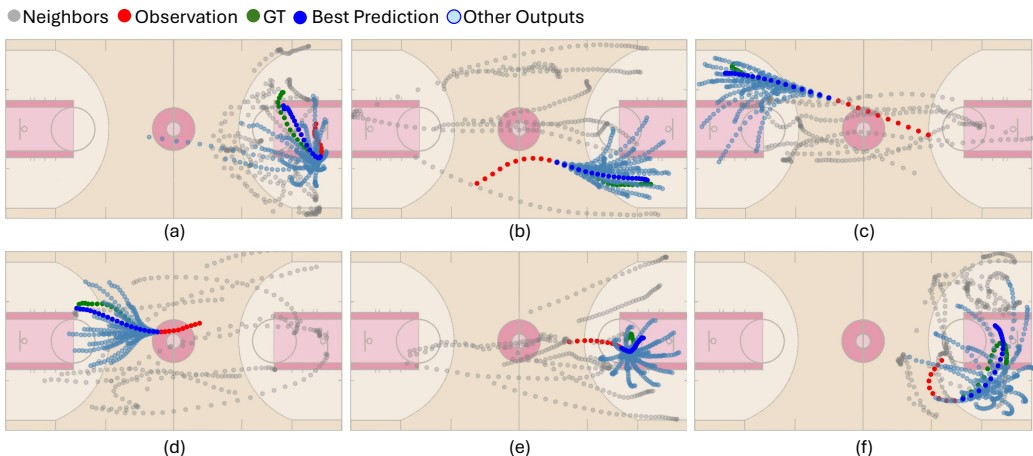

Figure 7: **Additional qualitative results of the NBA-Full with multimodal outputs**.

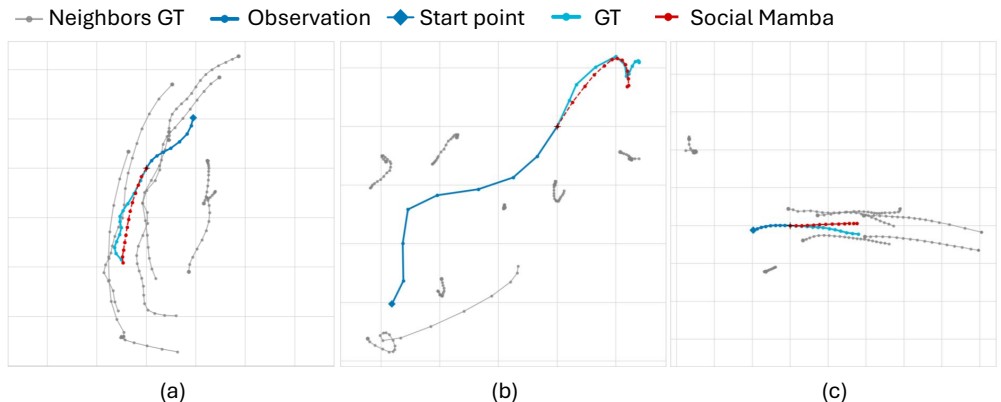

Figure 8: **Additional qualitative results on JRDB.** (a) Crowded environment. (b) Non-linear trajectory. (c) Failure case.

such as orientation or semantic roles (e.g., a goalie in sports). Future work could explore learnable or semantic-based ordering mechanisms. Second, our evaluations focus on pedestrian and sports datasets; extending to heterogeneous agents such as vehicles and cyclists would broaden applicability. Finally, while efficient, our current design predicts trajectories primarily for a single ego-agent. Extending to consistent, joint multi-agent prediction is an important and challenging direction. We hope Social-Mamba can serve as a strong foundation for these efforts.

TODO:Social grid

## A.8 LLM USAGE STATEMENT

We utilized a large language model (LLM) as a writing assistant to aid in the preparation of this manuscript. The LLM's role was strictly limited to improving the clarity, conciseness, and narrative flow of author-written text. This included tasks such as rephrasing sentences for better readability, suggesting alternative phrasings for technical descriptions, and ensuring a consistent tone. All scientific contributions, including the core ideas, experimental design, results, and analyses, were conceived and articulated by the human authors. In accordance with ICLR policy, the authors have meticulously reviewed all final text and take full responsibility for the content and integrity of this submission.

