# OpenReview forum: "Social-Mamba: Efficient Human Trajectory Forecasting with State-Space Models"
_ICLR.cc/2026/Conference — Submitted to ICLR 2026_

### Official Review · Reviewer_4div · 2025-10-24

**Soundness:** 2
**Presentation:** 2
**Contribution:** 2
**Rating:** 4
**Confidence:** 3

**Summary:**

This paper introduces Social-Mamba, a novel and efficient architecture for human trajectory forecasting that replaces quadratic-cost attention mechanisms with a Mamba-based state-space model to achieve linear-time complexity while maintaining state-of-the-art accuracy. Its core contributions include the Cycle Mamba block, which enables continuous bidirectional information flow for richer context modeling, and a structured social triplet factorization that decomposes interactions into temporal, egocentric, and goal-centric scans processed via semantically ordered sequences on an ego-centric grid. Extensive experiments on benchmarks demonstrate that Social-Mamba outperforms existing methods in accuracy while significantly improving parameter and computational efficiency.

**Strengths:**

1. The paper is logically structured and visually coherent, featuring well-designed figures, concise tables, and fluent writing. The clarity of presentation allows readers to grasp the methodology and results with ease.

2. The experimental analysis is thorough and well-executed.

**Weaknesses:**

1. **Novelty**: Mamba has been available for more than two years, and a large number of studies have already explored its application in various domains. Even within the trajectory forecasting field for both vehicles and humans, several works [1,2,3] have investigated the use of Mamba. Therefore, simply replacing the attention mechanism with Mamba for efficiency improvement is not sufficiently convincing or novel.

2. **efficiency**: Reporting only the number of parameters and GFLOPs is not enough to demonstrate the efficiency of the proposed method. A more comprehensive evaluation—including inference time and memory usage compared with other approaches—is necessary to substantiate the efficiency claims.

**References**:

[1] Trajectory Mamba: Efficient Attention-Mamba Forecasting Model Based on Selective SSM, CVPR 2025.

[2] DeMo: Decoupling Motion Forecasting into Directional Intentions and Dynamic States, NeurIPS 2024.

[3] MambaPTP: Exploring the Potential of Mamba for Pedestrian Trajectory Prediction, IEEE Transactions on Circuits and Systems for Video Technology.

**Questions:**

See  weaknesses.

---

> ### Author Response · Authors · 2025-11-19
>
> Thank you for your valuable and insightful feedback. We have addressed your concerns as follows.
>
> # Response to Weakness 1: Novelty and Additional References.
>
> We thank the reviewer for this constructive feedback. Although Mamba has achieved success in many domains, existing works for human trajectory prediction rely on hybrid methods, such as attention or attention-like Mamba for modeling social interaction (already discussed in the Introduction (lines 47~89)). The structural mismatch between 1D sequential SSMs and unstructured 2D social dynamics remains underexplored.
>
> First, we sincerely appreciate the additional references. We have incorporated a discussion of them into our revised manuscript (Introduction, lines 52, 85~88) to position our work better:
>
> 1. **Reference [1]:** reformulating the attention by Mamba-attention already discussed in line 84.
> 2. **Reference [2]:** this is a hybrid model that retains various attention modules. Our goal, conversely, is to fully leverage the linear-complexity advantage of SSMs by building a pure-Mamba architecture that eliminates the quadratic bottleneck entirely.
> 3. **Reference [3] (MambaPTP)**: We regard MambaPTP as a closely related parallel work (released after our ICLR submission). While it shares our motivation for using Mamba, it primarily operates by substituting for the attention mechanism within a standard decoder architecture to achieve social interaction. In contrast, our work focuses on reformulating the social scanning strategy itself. MambaPTP treats neighbors as a generic sequence, whereas our method imposes a semantic structure to resolve the spatial-temporal mismatch.
>
> We respectfully clarify that our contribution is not merely "being the first to use Mamba" or "replacing attention." Our novelty lies in **architecting a solution that makes social interactions compatible with sequential SSMs**, achieved through two key innovations:
>
> 1. **Social Triplet Factorization:** rather than a generic scan, we insert ego and goal tokens into neighbors’ sequence. This decomposes complex interactions into three semantic sequential processes (Temporal, Egocentric, and Goal-centric), effectively "linearizing" the unstructured social relationship.
> 2. **Cycle Mamba (CM) Block:** standard bidirectional Mambas (as used in prior works) suffer from information isolation between forward and backward passes. Our CM block introduces a continuous bidirectional flow, enabling "state-level fusion" that significantly improves continuity and efficiency compared to standard replacement methods.
>
> We believe these contributions represent a fundamental methodological advancement in how SSMs can be applied to multi-agent problems, going well beyond simple module replacement.
>
> We have **cited all the missing references and added the discussion** in our introduction, marked in blue.
>
> # Response to Weakness 2: Efficiency.
>
> We have expanded our efficiency analysis to provide a comprehensive evaluation. We measured inference time and model memory usage (excluding data) with a batch size of 1 to simulate a real-time application on NBA. Furthermore, we evaluated the GPU cost during training with a batch size of 128. All metrics were obtained on a single NVIDIA A100 GPU.
>
> The results in the table below confirm that the proposed Cycle Mamba block is highly efficient at the component level, yielding the **lowest memory usage and fastest inference time** compared to both MHSA and standard Bidirectional Mamba.
>
> | Model | Inference time (ms) | Model memory usage (MB) | Training memory usage (MB) |
> | :--- | :---: | :---: | :---: |
> | MHSA | 4.4 | 8.7 | 14356|
> | Bidirectional Mamba | 5.1 | 9.1 | 9422|
> | Cycle Mamba (ours) | **3.4** | **7.3** | **9375**|
>
> To further examine system-level efficiency, we compared our model against two Transformer-based baselines in the table below. We observe that the lightweight Social-Transmotion model achieves faster inference time; this is expected behavior attributable to the Transformer's inherent advantage in parallel computation for short sequences on GPUs. However, Social-Mamba remains extremely lightweight and is orders of magnitude faster than large-scale pretraining models like Multi-Transmotion. Crucially, Social-Mamba achieves the **lowest memory footprint** among all compared models, offering a superior balance of accuracy and resource efficiency for edge deployment.
>
> | Model | Inference time (ms) | Model memory usage (MB)
> | :--- | :---: | :---: |
> | Social-Transmotion | **1.8** | 7.6 |
> | Multi-Transmotion | 7.3 | 21.83 |
> | Social-Mamba (ours) | 3.4 | **7.3** |
>
>
> We have included these additional experiments and detailed discussions in Appendix A.3, marked in blue.

---

> > ### Comment · Reviewer_4div · 2025-11-27
> >
> > Thank you for the detailed response and the effort put into the rebuttal. My concerns have been addressed, and I will raise my score to 6. Since I am not very familiar with Human Trajectory Forecasting, I will lower my confidence level to 2. Overall, I have a positive assessment of this work.

---

### Official Review · Reviewer_EXmW · 2025-10-27

**Soundness:** 3
**Presentation:** 2
**Contribution:** 2
**Rating:** 4
**Confidence:** 4

**Summary:**

This paper proposes a Mamba-based trajectory prediction framework that reformulates social interactions as structured sequential processes. The framework is built on Cycle Mamba blocks to enable continuous bidirectional information flow, and introduces social triplet factorization to decompose interactions into temporal, egocentric, and goal-centric scans, which are finally integrated through a learnable social gate and global scan.

**Strengths:**

1. This paper presents a novel attempt to use sequential state-space models to model unstructured social interaction.
2. The proposed social triplet factorization provides an interpretable and modular approach to capture different aspects of social interaction.
3. The method achieves state-of-the-art performance on multiple benchmarks while maintaining high parameter efficiency and computational scalability. The comprehensive ablation studies demonstrate the effectiveness of each architectural design, further proving its robustness.

**Weaknesses:**

1. The strategy of ordering neighbors solely based on their distance to the target agent is not very sound. For example, a farther neighbor in front of the agent, who may potentially encounter the target, could have a stronger social influence than a closer neighbor moving away in the opposite direction. The social grid preparation process could therefore be designed more reasonably by considering more factors rather than distance alone.
2. The writing and presentation could be improved. For example, the complete output $O_{\mathrm{cycle}}$ in Line 223 does not appear in prior equations and lacks explanations. In addition, Eq.13 contains typographical errors that influence readability.
3. The key equations and variables need clearer explanations. Eq.13 and Eq.14 are presented without sufficient clarification of the underlying operations and intermediate variables. In particular, symbols such as $z_{e,t}^"$ are not explicitly defined, which makes it challenging to fully understand the computational flow and the overall method.

**Questions:**

1. In the social grid preparation process, the neighbors are sorted solely based on their distance to the target agent. Have the authors considered other factors in this process? If not, why it is rational to just use the distance?
2. What is the meaning of the "complete output $O_{\mathrm{cycle}}$ " in Line 223?
3. In Eq.13 and Eq.14, what are the precise definitions of the intermediate variables such as $z_{e,t}^"$?
4. How does the proposed framework produce multi-modal trajectory predictions?

---

> ### Author Response · Authors · 2025-11-19
>
> We sincerely appreciate your insightful feedback. We have addressed the weakness and questions as follows.
>
> # Response to Weakness 1  and Question 1: Sorting Strategies for Social Grid.
>
> We appreciate the reviewer's excellent question regarding the potential limitations of our distance-based ordering heuristic. To rigorously test the robustness of the Ego-centric Social Grid, we conducted an ablation study using three alternative neighbor arrangement strategies designed to isolate key social factors as follows.
>
> 1. **Collision risk:** sort neighbors by ascending Time-to-Collision (TTC). For approaching agents, TTC is approximated as $\tau= \frac{\left\|| p_{rel}\right\||^{2}}{p_{rel} ⋅v_{rel}}$, where $v_{rel}$ is the relative speed between the ego agent and a neighbor, and $p_{rel}$ is the relative position between the ego agent and a neighbor. Agents not on a collision course are assigned infinite TTC and sorted last.
>
> 2. **Goal alignment:** sort neighbors by descending cosine similarity between the neighbor's velocity $v_{neighbor}$ and the relative position vector pointing to the ego ($p_{ego} - p_{neighbor}$). This prioritizes neighbors explicitly moving toward the ego.
> 3. **Proximity:** apply a spatial filter that retains all agents within a relevance radius (e.g., \$10m$\) but imposes no specific order (random permutation). This isolates the impact of neighbor selection versus neighbor ordering.
>
> The ablation results in the table below on the NBA dataset reveal that all tested social heuristics, including the alternative sorting methods, yield similar ADE. This empirical evidence demonstrates that our Social-Mamba architecture is remarkably **insensitive** to the specific sequential ordering of neighbors within the Ego-centric Grid. This robustness is achieved by design: while the grid provides the necessary 1D sequential input structure for the Mamba scan, the subsequent Social Triplet Factorization and final fusion stage aggregate the information through a summation process, a mechanism that is theoretically permutation-invariant. The global interaction effectively learns to extract semantic features regardless of the heuristic used.
>
>
> | Strategy | minADE |
> | :--- | :---: |
> | Collision risk | 0.73 |
> | Goal alignment | 0.73 |
> | Proximity | 0.72 |
> | Distance | 0.72 |
>
> Thus, although proximity serves as a simple and efficient heuristic for ordering, the core architecture ensures that the final prediction relies on the content of the neighbor set, not the spatial rigidity of the sequence. That is, the global interaction can capture social relationships without relying on distance-based sorting.
>
> We have included these additional discussions and addressed the limitations in Appendix A.4, marked in blue.
>
> # Response to Weaknesses 2 and 3, and Questions 2 and 3: Clarification of notations.
> $O_{cycle}$ denotes the complete output sequence of the Cycle Mamba block. This output is generated following the SSM recurrence and subsequent linear projection layers within the Cycle Mamba block. We have updated Figure 2 and the accompanying text for improved clarity.
>
> In Equation (13), $z_{e,t}^{'}$ represents the ego index of $Z^{'E}$ along the time. In (14), $z_{e,t}^{''}$ in (14) denotes the ego index of $Z^{'G}$ along the time.
>
> We have clarified the ambiguous notations in the latest draft to improve presentation, marked in blue.
>
> # Response to Question 4: Clarification of Multi-modal Trajectory Predictions.
> The multi-modal predictions are generated by $K$ MLP heads in the decoder. We have added the descriptions in lines (332~337) to clarify.

---

### Official Review · Reviewer_E6p3 · 2025-11-01

**Soundness:** 3
**Presentation:** 2
**Contribution:** 2
**Rating:** 4
**Confidence:** 3

**Summary:**

This paper presents Social-Mamba, a trajectory forecasting architecture that adapts selective state-space models (Mamba) to model social interactions. The main technical contributions are (i) the Cycle Mamba block, a bidirectional SSM that concatenates reversed and forward sequences to enable continuous hidden-state flow, and (ii) a social triplet factorization that decomposes social interactions into temporal, egocentric, and goal-centric scans, fused via a learnable gating mechanism and a global scan. The method is evaluated across diverse benchmarks and achieves state-of-the-art performance while demonstrating notable parameter efficiency and computational scalability.

**Strengths:**

1. **Novel adaptation of SSMs to model social interactions.** The paper offers a thoughtful and original approach to use sequential state-space models to model unstructured social interaction. The Cycle Mamba design is conceptually simple yet effective: it provides a principled way to inject “future” context into forward processing while preserving Mamba’s efficiency advantages.
2. **Interpretable factorization of social interactions.** The social triplet factorization offers a modular and semantically meaningful decomposition of social interactions, which improves interpretability and enables targeted ablations to assess each component’s contribution.
3. **Strong performance and efficiency.** Social-Mamba achieves state-of-the-art performance across diverse benchmarks while using substantially fewer parameters and GFLOPs than several transformer-based baselines. The presented ablation studies further support the robustness of the architectural choices.

**Weaknesses:**

1. **Presentation and notation need improvement.** Several notational inconsistencies and typos make the technical flow harder to follow. For example, the "complete output $O_{\mathrm{cycle}}$" referenced around Line 223 is introduced without prior formal definition; Eq. 13 and Eq. 14 use variables (e.g., $z_{e,t}^"$) that are not clearly defined in the main text. The authors should carefully check these issues for better readability.
2. **Neighbor ordering by Euclidean distance is simplistic.** Sorting neighbors solely by Euclidean distance (Eq.9) is a weak heuristic for social importance. The authors should either justify this choice theoretically/empirically or provide experiments that compare alternative motion-aware ordering or weighting schemes.
3. **Incomplete description of multi-modal predictions.** Section 4.3 states the model predicts $K$ trajectories and trains with a best-of-K loss, but it is unclear how the $K$ modes are generated at inference. Clarifying this is important for understanding how diversity is achieved and evaluated.

**Questions:**

1. The social grid permutation $\pi$ (Eq.9) uses distance at $T_{\mathrm{obs}}$. Have the authors tried ordering by alternative criteria? If so, please report results; if not, please discuss why distance is sufficient and outline how the model might incorporate richer ordering.
2. How are the $K$ predicted trajectories generated? Is the decoder trained to produce $K$ hypotheses deterministically, or are the hypotheses obtained via sampling from a latent distribution?
3. What are the precise definitions for variables in Eq.13 and Eq.14? What are the meaning of the calculation of each term in these two equations?

---

> ### Author Response · Authors · 2025-11-19
>
> We sincerely appreciate your valuable feedback. We have addressed the weakness and questions as follows.
>
> # Response to Weakness 1 and Question 3: Clarifying Notations and Improving Presentation.
> Thanks for the careful check of our work. $O_{cycle}$ refers to the output of the cycle mamba block after the projection, after the SSM. We clarify both in the text and Figure 2.  $z_{e,t}^{'}$ in (13) is the ego index of $Z^{'E}$ along the time.
> Similarly, $z_{e,t}^{''}$ in (14) is the ego index of $Z^{'G}$ along the time.
> We have clarified the ambiguous notations in the latest draft, marked in blue.
>
> # Response to Weakness 2 and Question 1: Impact of Ordering.
>
> We appreciate the reviewer's query regarding the potential limitations of our distance-based ordering, as the sensitivity of any imposed structure is a valid concern. To rigorously examine the robustness of the Ego-centric Social Grid, we conducted an ablation study using three alternative neighbor arrangement strategies as follows.
>
> 1. **Collision risk:** sort neighbors by ascending Time-to-Collision (TTC). For approaching agents, TTC is approximated as $\tau= \frac{\left\|| p_{rel}\right\||^{2}}{p_{rel} ⋅v_{rel}}$, where $v_{rel}$ is the relative speed between the ego agent and a neighbor, and $p_{rel}$ is the relative position between the ego agent and a neighbor. Agents not on a collision course are assigned infinite TTC and sorted last.
>
> 2. **Goal alignment:** sort neighbors by descending cosine similarity between the neighbor's velocity $v_{neighbor}$ and the relative position vector pointing to the ego ($p_{ego} - p_{neighbor}$). This prioritizes neighbors explicitly moving toward the ego.
> 3. **Proximity:** apply a spatial filter that retains all agents within a relevance radius (e.g., \$10m$\) but imposes no specific order (random permutation). This isolates the impact of neighbor selection versus neighbor ordering.
>
> The ablation results on the NBA dataset in the table below reveal that all tested heuristics yield **comparable ADE**. This empirical evidence demonstrates that our Social-Mamba architecture is **largely insensitive to the specific sequential ordering** of neighbors within the Ego-centric Grid. This inherent robustness is by design: it aligns with the structure of our Social Triplet module, which aggregates and fuses the information from the three parallel scans via summation, a process that is theoretically permutation-invariant. That is, the global interaction works on different reasonable heuristics.
>
> | Strategy | minADE |
> | :--- | :---: |
> | Collision risk | 0.73 |
> | Goal alignment | 0.73 |
> | Proximity | 0.72 |
> | Distance | 0.72 |
>
> We have included these additional discussions and addressed the limitations in Appendix A.4, marked in blue.
>
> # Response to Weakness 3 and Question 2: Clarifying Multi-modal Predictions.
> The multi-modal predictions are generated by $K$ MLP heads. The best-of-K loss is calculated by selecting the best prediction. We have added the description around $\hat{Y}_e = \text{Decoder}(\mathbf{Z}^{\text{final}}_e)$ (17) to help understanding.

---

### Official Review · Reviewer_U1cm · 2025-11-04

**Soundness:** 4
**Presentation:** 3
**Contribution:** 4
**Rating:** 8
**Confidence:** 3

**Summary:**

The paper introduces Social-Mamba, a novel architecture for human trajectory forecasting designed to achieve high accuracy while providing superior computational efficiency and scalability compared to conventional methods.  The core innovations include the Cycle Mamba (CM) block, a bidirectional SSM module that ensures continuous information flow; the Ego-centric Social Grid, which resolves the inherent ordering problem of sequential models by sorting neighbors based on distance to the ego agent; and Social Triplet Factorization, which decomposes interactions into temporal, ego-centric, and goal-centric sequential scans aggregated via a learnable social gate. The architecture achieves state-of-the-art accuracy across five benchmarks while demonstrating superior parameter efficiency and computational scalability.

**Strengths:**

Social-Mamba successfully addresses the critical drawback of attention mechanisms (quadratic computational cost) by utilizing the Mamba framework, offering superior parameter efficiency and computational scalability

The paper adapts the inherently sequential design of SSMs to model complex, unstructured social interactions. This is achieved through the introduction of the Cycle Mamba block, which facilitates continuous bidirectional information flow, and the Ego-centric Social Grid, which imposes a meaningful ordering necessary for sequential processing of spatial data

Despite prioritizing efficiency, Social-Mamba attains state-of-the-art accuracy on five diverse trajectory forecasting benchmarks

The Social Triplet Factorization technique effectively breaks down complex social dynamics into separate, purposeful sequential scans (temporal, ego-centric, and goal-centric), which ablation studies demonstrated yield the best overall performance

**Weaknesses:**

While the Ego-centric Social Grid resolves the ordering problem inherent to SSMs, the performance may rely heavily on the quality and consistency of the distance-based ordering. This imposed structure might potentially constrain the natural, unstructured dependencies that attention mechanisms were designed to capture, although the performance suggests this constraint is effectively managed.

The design successfully introduces the Cycle Mamba block for continuous bidirectional flow. While this is innovative, ensuring that the imposed sequential structure does not inadvertently introduce causal or temporal biases during the bidirectional scan remains a potential issue not fully detailed in the summary.

**Questions:**

How sensitive is the performance of the model to alternative ordering strategies within the Ego-centric Social Grid (e.g., ordering by predicted collision risk or goal alignment rather than distance)?

Could the authors elaborate on the specific limitations and future directions discussed in the appendix, particularly concerning scenarios where the Ego-centric Social Grid might inadequately represent crucial, non-distance-dependent social interactions?

---

> ### Author Response · Authors · 2025-11-19
>
> We sincerely thank you for your insightful feedback. We have addressed your concerns as follows.
>
> # Response to Weakness 1, Question 1 and 2: Performance and Sensitivity of Ordering Strategy.
>
> **(W1, Q1)** To examine the sensitivity of the social grid, we have designed three additional strategies to arrange neighbors as follows.
>
> 1. **Collision risk:** sort neighbors by ascending Time-to-Collision (TTC). For approaching agents, TTC is approximated as $\tau= \frac{\left\|| p_{rel}\right\||^{2}}{p_{rel} ⋅v_{rel}}$, where $v_{rel}$ is the relative speed between the ego agent and a neighbor, and $p_{rel}$ is the relative position between the ego agent and a neighbor. Agents not on a collision course are assigned infinite TTC and sorted last.
>
> 2. **Goal alignment:** sort neighbors by descending cosine similarity between the neighbor's velocity $v_{neighbor}$ and the relative position vector pointing to the ego ($p_{ego} - p_{neighbor}$). This prioritizes neighbors explicitly moving toward the ego.
> 3. **Proximity:** apply a spatial filter that retains all agents within a relevance radius (e.g., \$10m$\) but imposes no specific order (random permutation). This isolates the impact of neighbor selection versus neighbor ordering.
>
> We present ablation results on the NBA dataset in the table below, where all heuristics yield **comparable ADE**. This empirical evidence reveals that our model is **largely insensitive to the specific order of neighbors** with a reasonable heuristic. This robustness aligns with the design of our Social Triplet module, which aggregates interactions via summation, a process that is theoretically permutation-invariant.
>
> | Strategy | minADE |
> | :--- | :---: |
> | Collision risk | 0.73 |
> | Goal alignment | 0.73 |
> | Proximity | 0.72 |
> | Distance | 0.72 |
>
>
> **(Q2)** The results also indicate that our model can learn from different reasonable heuristics. The heuristics do not need to be complicated.  The limitation is that the selected heuristic must rely on observable information in a real-world system.
>
> We have added additional discussion and limitations in Appendix A.4 marked in blue.
>
> # Response to Weakness 2: Temporal Bias and Causal Integrity of the Cycle Mamba Block.
> We appreciate this thoughtful observation. Cycle Mamba (CM) introduces a structural asymmetry where the forward pass is conditioned on the backward pass. We clarify that this is not an inadvertent bias, but a **beneficial inductive bias designed to model "hindsight."** We validate this claim both theoretically and empirically:
> 1. **Theoretical—encoder context:** the CM block operates as an encoder on observed history. In this non-causal context, accessing the "future" of the sequence is standard practice for building global context. Unlike standard bidirectional models that process directions in isolation (contextual isolation), our design uses the backward pass to initialize the forward pass (state-level fusion). This mimics human reasoning: interpreting early ambiguous actions (e.g., a turn) using knowledge of the final destination.
>
> 2. **Empirical—comparison to "unbiased" attention:** to ensure this structural bias is not harmful, we compared Cycle Mamba against Multi-Head Self-Attention (MHSA) on NBA dataset in the table below. MHSA is inherently permutation-invariant and lacks sequential bias, offering a pure global view of the context.
>
> | Strategy | minADE/minFDE |
> | :--- | :---: |
> | MHSA | 0.74/0.95 |
> | Cycle Mamba | 0.72/0.92 |
>
>
> The results demonstrate that the imposed sequential structure neither imposes a limitation nor introduces errors. Instead, it serves as an **effective inductive bias**, structuring the bidirectional information flow more efficiently than the unbiased view of standard attention.

---

### Author Response · Authors · 2025-11-26

Dear Reviewers,

This is a friendly reminder that our rebuttal has been submitted for your consideration.

We sincerely hope our responses address your initial concerns. We are ready and happy to provide additional clarification if any questions remain after reviewing our response.

Thank you once more for your valuable feedback and commitment to the review process.

---

### Author Response · Authors · 2025-12-01
**Review and Rebuttal Summary**

Dear AC,

Thank you for your time and effort in managing our submission and the reviewer discussion. To support your assessment, we provide a brief summary of the reviewers’ comments, our responses, and the corresponding revisions made to the manuscript.

**Strengths:**

**Novel Design:** Reviewers acknowledge our core method design as a significant contribution. It is described as a "thoughtful and original approach" (Reviewer E6p3) and a "novel attempt to use sequential state-space models to model unstructured social interaction" (Reviewer EXmW). Reviewer U1cm highlighted that the Cycle Mamba block "facilitates continuous bidirectional information flow," while Reviewer E6p3 noted the design is "conceptually simple yet effective."

**Strong Performance:** The overall performance is widely recognized. Reviewers noted the method achieves "state-of-the-art accuracy across five benchmarks" (Reviewer U1cm) and "outperforms existing methods in accuracy" (Reviewer 4div).

**Efficiency:** A key strength highlighted is that our method offers "superior parameter efficiency and computational scalability" (Reviewer U1cm) and uses "substantially fewer parameters and GFLOPs than several transformer-based baselines" (Reviewer E6p3).

**Interpretable Design:** The Social Triplet Factorization is praised for offering a "modular and semantically meaningful decomposition" (Reviewer E6p3) that "improves interpretability" (Reviewer EXmW).


**Weakness and responses:**

**Critiques:** The primary technical concern raised by Reviewers U1cm, E6p3, and EXmW was regarding the Ego-centric Social Grid, specifically whether ordering neighbors solely by distance is a sufficient heuristic or if it limits the model's ability to capture complex social dynamics. Additionally, Reviewer 4div questioned the novelty regarding the general use of Mamba and requested more detailed efficiency metrics (inference time/memory).

**Our response:**

**Robustness of Ordering Strategy:** In our rebuttal, we addressed the concerns regarding neighbor ordering by conducting extensive ablation studies using three alternative strategies: Collision Risk, Goal Alignment, and Proximity. The results (added to Appendix A.4) demonstrate that our model yields comparable ADE across all heuristics. This empirically proves that our architecture is insensitive to specific neighbor ordering, validating our design choice where the Social Triplet module aggregates interactions in a permutation-invariant manner.

**Novelty and Positioning:** We clarified to Reviewer 4div that, unlike concurrent hybrid works that simply replace attention modules, our contribution lies in architecting a solution (via Social Triplet Factorization and Cycle Mamba) that fundamentally makes unstructured social interactions compatible with sequential SSMs. We have included and discussed the suggested references in the Introduction.

**Efficiency Analysis:** We provided the requested system-level efficiency breakdown. Our new experiments (added to Appendix A.3) show that Cycle Mamba achieves faster inference times (3.4ms) and lower memory usage (7.3MB) compared to both MHSA and standard Bidirectional Mamba, substantiating our efficiency claims.

**Reviewer 4div already responded that we have addressed the concerns about Novelty and Efficiency.**

**Clarifications:** The requests for clarification regarding notation (e.g., $O_{cycle}$, Eq. 13/14) and multi-modal prediction generation raised by Reviewers E6p3 and EXmW have been addressed. We have refined the definitions and mathematical descriptions in the revised manuscript to improve readability and presentation.

**Summary:** We have conducted all suggested experiments, including robustness checks for neighbor ordering and detailed efficiency benchmarks. These results strongly support our initial model design and have been incorporated into the revised manuscript along with improved notation. We believe these additions address the reviewers' concerns regarding the robustness of the grid heuristic and the depth of the efficiency evaluation. We appreciate all reviewers for their constructive feedback, which has helped us significantly improve the paper. Lastly, we again thank the AC for your time and effort.

---

### Meta-Review · Area_Chair_jpWN · 2025-12-25

**Summary:**

Reviewers praised the adaptation of state-space models to capture social interactions, the interpretable factorization of interaction components, and the computational efficiency of the proposed approach. However, reviewers also raised multiple concerns that limit confidence in the contribution. Several questioned the sensitivity of the method to alternative ordering strategies and requested clearer explanations about the model itself (e.g., asking for a clearer mathematical formulation). One reviewer also explicitly questioned the novelty of the approach. Most reviewers also indicated unfamiliarity with the broader field or aspects of the work with confidences of 3 or below.

In addition to the points raised by reviewers, a core concern from my own assessment of the paper is the choice and suitability of datasets used for evaluation. The NBA dataset employed in the paper does not have a license on its publicly hosted repository, raising questions about reproducibility and responsible data usage. Moreover, the SDD and JRDB datasets are small-scale and highly saturated, having been extensively studied for over a decade. As a result, the reported performance improvements over baselines are extremely small, on the order of 1–2 cm across multiple tables (Tables 1, 3, 4, 6, 7), which makes it difficult to meaningfully assess the practical improvement in trajectory quality from the proposed model.

Finally, there are minor presentation issues, including a “TODO” left in Appendix A.7, which further suggests that the paper may not yet be in a final state.

Overall, while the paper contains promising components, the combination of concerns around novelty, clarity, evaluation strength, dataset suitability, and reviewer confidence make it difficult to recommend acceptance at this time.

**Reviewer Concerns:**

I believe the authors did a good job of addressing most reviewers' questions and weaknesses. However, I share the core novelty concern of reviewer 4div; it is something that additional discussions in the related work and introduction do not entirely address (ideally, these methods should be included in the experiments to provide readers with additional information about how Social-Mamba fits within the literature and works applying a similar architecture).

SAC agrees with AC on core novelty concern.

**Reviewer Scores:**

I do not think that reviewer U1cm would have changed their score.

It is difficult to ascertain if reviewers E6p3 and EXmW would have changed their scores as they had remaining questions about the core methodology due to certain aspects of the writing and key equations being difficult to parse (and it is difficult to know if their deeper understanding of the method would make them more positive of the approach or more certain of their core concerns).

4div indicated that they would increase their score to a 6 (and simultaneously lower their confidence to a 2). However, I struggle to envision the same outcome if 4div was a more confident reviewer (as I view their core novelty concern to remain generally unaddressed).

---

### Decision · Program_Chairs · 2026-01-26

Reject